# Impact of Marine Atmospheric Corrosion on the Microstructure and Tensile Properties of 7075 High-Strength Aluminum Alloy

**DOI:** 10.3390/ma16062396

**Published:** 2023-03-16

**Authors:** Lin Xiang, Jianquan Tao, Xiangsheng Xia, Zude Zhao, Qiang Chen, Yan Su, Shuxin Chai, Zhongyan Zheng, Jipeng Sun

**Affiliations:** Southwest Technology and Engineering Research Institute, Chongqing 400039, China2009chenqiang@163.com (Q.C.);

**Keywords:** 7075 alloy, corrosion crack, intergranular corrosion, microstructure, tensile properties

## Abstract

This study aimed to investigate the impact of corrosion on the microstructure and tensile properties of 7075 high-strength alloy. It involved outdoor exposure tests in an actual marine atmospheric environment in Wanning, Hainan Province. The results showed that the 7075 alloy was corroded rapidly in the marine atmospheric environment, and corrosion pits and intergranular cracks were generated. The intergranular cracks were extended along the grain boundary during corrosion, leading to the exfoliation of the matrix. The cause for the intergranular corrosion was discussed based on the microstructure characteristics of 7075 alloy. The tensile properties of the 7075 alloy gradually deteriorated with the increase of exposure time in the marine atmospheric environment. The ultimate strength and elongation were decreased by about 3.2% and 58.3%, respectively, after 12 months of outdoor exposure.

## 1. Introduction

Aluminum and its alloys have been widely used in the engineering field, especially in the aerospace, aviation, and automotive industries [1,2,3,4]. Among aluminum alloys, 7075 alloy, which is a typical 7XXX series aluminum alloy, has attracted the attention of scientists because of its excellent mechanical properties [5,6,7,8]. When aluminum alloy components are exposed to marine atmospheric environments, they are corroded due to hostile environments, such as high salt spray and high humidity, resulting in the deterioration of mechanical properties [9,10,11]. Therefore, it is important to understand the effect of an actual marine environment on the microstructure and mechanical properties of the alloy.

Some studies to date explored the corrosion behavior of 7075 alloy [12,13,14,15,16]. Marta Orłowska et al. studied the impact of heat treatment on the mechanical properties and corrosion resistance of ultra-fine-grained 7075 alloy and found that plastic deformation increased susceptibility to corrosion due to an increase in grain boundaries [12]. Liwei Wang et al. studied the corrosion evolution of high-strength aluminum alloys in a solution and a thin electrolyte layer containing Cl^−^ and found that large corrosion cavities with limited underlying microcracks were formed [14]. Xiaohang Liu et al. studied the effect of nitrate on the corrosion behavior of 7075-T651 aluminum alloy in an acidic NaCl solution and revealed that the nitrate promoted the pitting corrosion once the stable pitting corrosion was initiated [16]. However, few studies have been conducted in an actual marine environment which reflect the corrosion behavior of alloy components during service. Therefore, in this study, outdoor exposure tests were conducted in the South China Sea. Then, the variation of microstructures and tensile properties were analyzed, and the corrosion mechanism was examined.

## 2. Experimental Section

According to previous study [17], 7075-T73 (over-aged) alloy may possess a lower strength and better corrosion resistance compared with 7075-T6 (peak-aged) alloy. Thus, a 7075-type aluminum alloy rolled plate with a heat-treatment status of T73 was used as the object in this study. Its chemical composition was analyzed via inductively coupled plasma atomic emission spectrometric method (OES8000), and the results are listed in Table 1. The outdoor exposure test in the marine atmospheric environment was carried out in Wanning, Hainan Province, China. A previous study indicated that the average temperature, relative humidity, and Cl^−^ deposition rate at Wanning were about 23.9 °C, 87.6%, and 14.5875 mg/(m^2^ d), respectively [18]. Two types of specimens were used: a plate with a size of 100 × 50 × 3 mm^3^ for observing the cross-section corrosion morphology and a dumbbell-shaped specimen with a test section diameter of 10 mm for testing the tensile properties, which were cut from rolled plate by wire-electrode cutting, and then they were polished to ensure the surfaces were smooth. All specimens were machined from the 7075 alloy rolled plate whose length was parallel to the extrusion direction. The specimens were placed in a rack that was fixed 45° to the south in an outdoor exposure field, as shown in Figure 1. The field was 350 m away from the ocean. After 1, 6, and 12 months of outdoor exposure, the specimens were used for observing the macroscopic morphology. After taking the photographs, the exposure test was continued. The total outdoor exposure time was 12 months. The sampling was performed every 6 months to observe the changes in weight loss, microstructure, and mechanical properties of the alloy. Three specimens were tested in each test, and their average value was shown in this study.

The weight loss was analyzed according to Chinese Standard 16545-2015 (*Corrosion of Metals and Alloys. Removal of Corrosion Products from Corrosion Test Specimens*). The cross-section morphology and microstructure of the specimen were observed using a DFC320-type optical microscope (Leica Microsystems, Wetzlar, Germany). The microstructure and fracture morphology of the specimen were observed using a JSM-6390A-type scanning electron microscope (JEOL, Tokyo, Japan). The secondary-phase composition was analyzed using a D/MAX-2200-PC-type X-ray diffractometer (Rigaku Corporation, Tokyo, Japan) with an X-ray source equipped with a Cu (Kα) target and detection wavelength λ of 1.5406 Ǻ. A Tecnai-F30-G2-type transmission electron microscope (Hillsboro, OR, USA) with an accelerating voltage of 300 kV was used to analyze the secondary phase characteristics of the alloy. A universal testing machine was employed to test the tensile properties at room temperature. The tensile rate was 2 mm/min. Three specimens were examined in each test, and the mean value was taken as the actual value.

## 3. Results and Discussion

### 3.1. Initial Microstructure of the 7075 Alloy

Figure 2 shows the original microstructure of the 7075 alloy. Figure 2a,b shows that the microstructure was principally composed of an α(Al) matrix and secondary phase particles. The matrix grains were in the shape of long strips. The precipitated secondary phase particles were not only distributed along the matrix grain boundary but also present inside the α-Al grains. The secondary phase particles were determined using XRD. The results are shown in Figure 2c. The secondary phase particles were principally composed of the MgZn_2_ phase. Further, the secondary phases also contained a small amount of Al_2_CuMg phase.

### 3.2. Corrosion Behavior of the 7075 Alloy

Figure 3 shows the macroscopic morphologies of the 7075 alloy after exposure to the marine atmospheric environment at different time points. The alloy was corroded in the marine atmospheric environment after only 1 month of exposure. As shown in Figure 3a, the surface of the specimen turned dark gray, and some black-dotted corrosion products were scattered. As shown in Figure 3b,c, the number of black products on the surface gradually increased with the increase of exposure time, indicating that the degree of corrosion of the alloy was higher. After exposure for 12 months, the black-dotted corrosion products were densely distributed on the surface of the specimen. However, the analysis of the average weight loss showed that the values after exposure for 6 months and 12 months were 0.0892 g and 0.1789 g, respectively. Correspondingly, the corrosion rates of the present alloy were 0.0447 g/(m^2^ d) and 0.0450 g/(m^2^ d), respectively. These result indicated that the weight loss increased with the increase of exposure time, but the corrosion rates did not change during exposure for 12 months. Our previous research indicated that the average weight losses of the 7085 alloy exposed for the same time in the same marine environment were 0.0231 g and 0.0294 g, respectively [19]. It was obvious that the 7075 alloy corroded faster.

The corrosion products of the 7075 alloy in the marine atmospheric environment were analyzed using XRD, as shown in Figure 4a. The corrosion products principally consisted of Al(OH)_3_ and AlCl_3_. EDS was executed to further analyze the corrosion product, as shown in Figure 4b. The EDS results indicated that the main elements of corrosion product were Al, O, and Fe, and the atomic percentage of Al and O was 1:2. This result also indicated that the corrosion product was not just Al(OH)_3_ or AlCl_3_. Rather, it was a mixture of both. Moreover, traces of Fe element were found in the corrosion product. According to previous reports [20,21,22], Fe-rich constituent particles existed in the 7075 alloy, which had a higher electrochemical potential than the Al matrix. In other words, the corrosion behavior preferentially occurred around Fe-rich constituent particles [20]. Figure 4b also shows the morphology of the corrosion products on the specimen surface, indicating that the specimen surface was closely coated by the dense corrosion products capable of protecting the matrix during exposure. Therefore, the corrosion rate of the present alloy was always similar after exposure for 6 months and 12 months.

Figure 5 shows the cross-sectional morphologies of the 7075 alloy after exposure to the marine atmospheric environment during different periods. After 6 and 12 months of exposure, corrosion pits were observed on the specimen surface (Figure 5b,c). Meanwhile, evident corrosion cracks were also observed in the alloy, and the cracks were developed parallel to the specimen surface. Further, the extension of cracks was critical during the exposure. After 12 months of exposure, the exfoliation of the matrix occurred. The local magnification of Figure 5c is shown in Figure 5d. The corrosion cracks were principally developed along the grain boundary with their texture elongated, showing typical features of intergranular corrosion (IGC). The aforementioned results showed that obvious IGC took place in the 7075 alloy in the marine atmospheric environment, and intergranular cracks were generated along the grain boundary of the matrix. The intergranular cracks were gradually extended, and the corrosion was intensified, leading to the exfoliation of the matrix. Many excellent studies have been conducted on the mechanism of exfoliation corrosion [23,24,25] and indicated that two conditions were responsible for exfoliation corrosion. One was prolonged matrix gains, and another was IGC. Obviously, in our work, both conditions for exfoliation corrosion could be satisfied. Therefore, exfoliation corrosion occurred in the 7075 alloy after exposure for 12 months in the marine atmospheric environment. In addition, the IGC depths of the specimens exposed for 6 months and 12 months were 44.5 μm and 53.6 μm, respectively. The increase of IGC depth from 6 months to 12 months was only 9.1 μm, indicating that intergranular cracks developed mainly along the grain boundary. In order to discuss the reason for IGC, the microcharacteristics of the 7075 alloy were analyzed.

### 3.3. Analysis of the Microcharacteristics of the 7075 Alloy

The characteristics of the secondary phase of the alloy were analyzed using transmission electron microscopy (TEM) to understand the IGC and exfoliation of the 7075 alloy. As shown in Figure 6a,b, the secondary phases of the 7075 alloy were principally of two types: the granular secondary phase that was continuously distributed along the grain boundary and the nano-sized secondary phase that was uniformly distributed in the grains. Both secondary phases were analyzed using the selected area electron diffraction method, and the results are shown in Figure 6c–e. The secondary phase along the grain boundary was composed of S(Al_2_CuMg) and stable η(MgZn_2_) phases, while the secondary phase in the grains was the metastable $\eta^{\prime }$(MgZn_2_) phase. These were consistent with the results of the XRD analysis. The reason for this phenomenon was that the precipitation sequence of the secondary phase during the aging treatment of 7XXX-type aluminum alloy was as follows: supersaturated solid solution → GP zone → metastable $\eta^{\prime }$ phase → stable η phase [26,27,28]. Among these, the GP zone and metastable $\eta^{\prime }$ phase were usually precipitated inside the α-Al crystalline grains, while the stable η phase was usually precipitated along the grain or subgrain boundaries [29].

Figure 7 shows the high-resolution images of the secondary phases. As shown in Figure 7a, the lattice parameters of the η phase were a = b = 0.522 nm and c = 0.857 nm, showing an incoherent relationship with the Al matrix. The lattice parameters of the metastable $\eta^{\prime }$ phase and S phases were a = b = 0.505 nm, c = 1.402 nm and a = 0.4 nm, b = 0.925 nm, c = 715 nm, respectively, as shown in Figure 7b,c.

The potential of the η(MgZn_2_) phase in the 7XXX-type aluminum alloy was lower than that of the Al matrix, and the values were −0.86 V and −0.68 V, respectively [30]. Therefore, the η(MgZn_2_) phase was usually consumed as the anode phase. The S(Al_2_CuMg) phase was rich in Cu and Mg, and this phase was the anode phase relative to the Al matrix in the marine atmospheric environment. The Mg element was selectively dissolved. The Cu-rich residual phase remained and was transformed into a cathode phase relative to the Al matrix, accelerating the corrosion of the Al matrix. Therefore, the corrosion mechanism of 7075 alloy in a humid and hot marine atmospheric environment could be speculated as follows: The η(MgZn_2_) and S phases along the grain boundary were dissolved as the anodes in the initial stage of corrosion, resulting in IGC, and the η phase, which was continuously distributed along the grain boundary, favored the development of intergranular cracks. The S phase was gradually transformed from the anode to the cathode phase with the dissolution of the Mg element in the S phase. At this time, the Al matrix in the vicinity of the S phase along the grain boundary was corroded to generate an Al(OH)_3_ product, which led to an “embedding effect” along the grain boundary. Further, as corrosion proceeded, the IGC and “embedding effect” were gradually intensified, while the strip-shaped grain structure could not transfer the growing direction of intergranular cracks. The intergranular cracks were always extended along the long side of the grain. As a result, the matrix was exfoliated. The corrosion processes sketch of the 7075 alloy’s exposure to the marine atmospheric environment is shown in Figure 8.

### 3.4. Deterioration of Tensile Properties of the 7075 Alloy

Figure 9 shows the changes in the mechanical properties of the 7075 alloy after exposure to the marine atmospheric environment at different time points. As shown in Figure 9a, the ultimate and yield strengths of the 7075 alloy decreased gradually during exposure. After 6 months of exposure to the marine environment, the ultimate and yield strengths of the 7075 alloy decreased from 611 and 556 to 605 and 551 MPa, respectively. After 12 months of exposure, the ultimate and yield strengths were further decreased to 591 and 547 MPa, respectively. Although the strength of the 7075 alloy declined gradually during exposure to the marine atmospheric environment, the decrease in strength was limited even after 12 months of exposure. The ultimate strength was decreased by only about 3.2% after 12 months of exposure. On the contrary, the elongation and reduction of the area of the 7075 alloy also declined during exposure (Figure 9b). Noticeably, the decrease in elongation and reduction of the area was obvious. After 6 months of exposure, the elongation and reduction of the area decreased from 12% and 15% to 8% and 12%, respectively. After 12 months of exposure, the elongation and reduction of the area decreased to 5% and 8%, respectively, and the decrease exceeded 45%. According to the analysis of corrosion behavior, corrosion pits and intergranular cracks were generated in the 7075 alloy in the humid and hot marine environment. These corrosion defects were a highly possible source for the generating of cracks during the tensile process, leading to the deterioration of alloy properties, especially plasticity. Figure 10a shows the morphology of the tensile fracture of the 7075 alloy after exposure to the marine atmospheric environment. Obvious corrosion pits were observed on the specimen surface. Meanwhile, the cross-sectional morphology of the tensile specimen was observed (Figure 10b). Besides the corrosion pits on the specimen surface, an obvious IGC phenomenon in the specimen was observed, confirming that corrosion pits and intergranular cracks were the dominant reasons for the deterioration of the mechanical properties of the alloy.

## 4. Conclusions

In this work, the corrosion behavior and deterioration rules of the mechanical properties of 7075 high-strength aluminum alloy in a marine atmospheric environment were studied. The findings of this study were as follows:

(1) The 7075 alloy was corroded rapidly in the marine atmospheric environment. Corrosion products Al(OH)_3_ and AlCl_3_ were generated on the alloy surface, and intergranular cracks were formed in the alloy. The degree of corrosion of the alloy was gradually intensified during exposure, and intergranular cracks were critically extended along the grain boundaries, leading to the exfoliation of the matrix.

(2) In the marine atmospheric environment, the η(MgZn_2_) and S phases along the grain boundary of the 7075 alloy were dissolved as the anodes, resulting in the IGC. With the consumption of the Mg element in the S phase, the Al matrix neighboring the S phase at the grain boundary was corroded to provide the Al(OH)_3_ product, which led to the “embedding effect” along the grain boundary and exfoliation of the matrix.

(3) In the marine atmospheric environment, the strength and plasticity of the 7075 alloy gradually declined during exposure, and the plasticity of the alloy was extremely sensitive to the corrosion pits and intergranular cracks. After 6 months of exposure, the elongation and reduction of area decreased from 12% and 15% to 8% and 12%, respectively. Increasing exposure time to 12 months, the plasticity of the alloy decreased by more than 45%.

## Figures and Tables

**Figure 1 materials-16-02396-f001:**
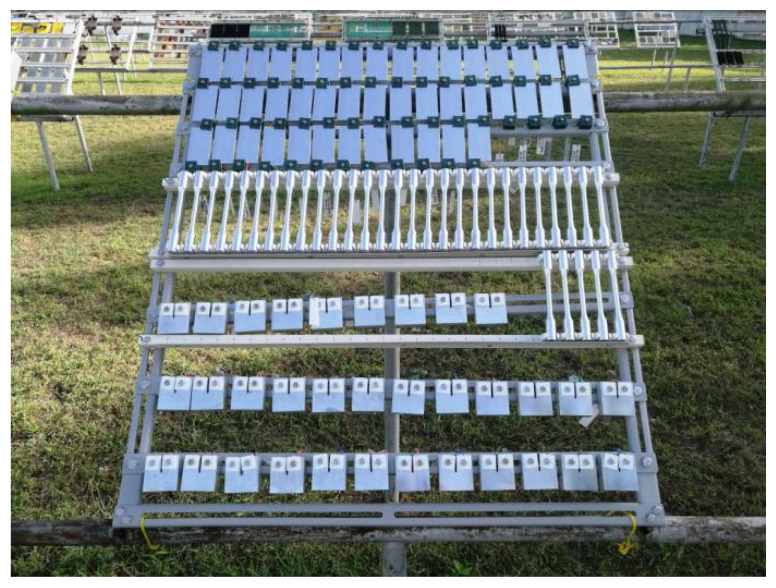
Outdoor exposure test of the 7075 alloy in the marine atmospheric environment.

**Figure 2 materials-16-02396-f002:**
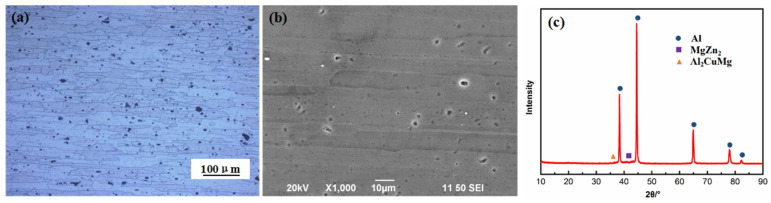
Original microstructure images of the 7075 alloy taken using (**a**) optical microscope, (**b**) SEM, and (**c**) XRD.

**Figure 3 materials-16-02396-f003:**
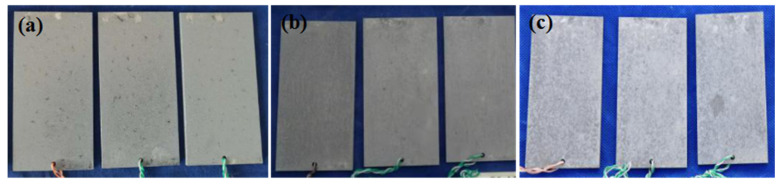
Macroscopic morphologies of the 7075 alloy after exposure at different time points in the marine atmospheric environment: (**a**) 1 month, (**b**) 6 months, and (**c**) 12 months.

**Figure 4 materials-16-02396-f004:**
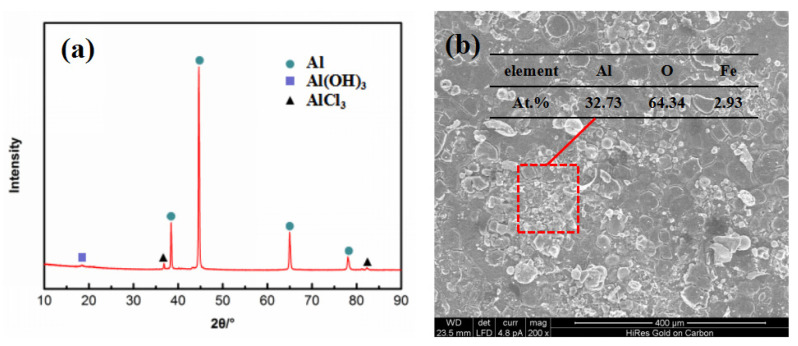
Images of corrosion products of the 7075 alloy after exposure for 6 months in the marine atmospheric environment: (**a**) XRD, (**b**) SEM morphology, and EDS result.

**Figure 5 materials-16-02396-f005:**
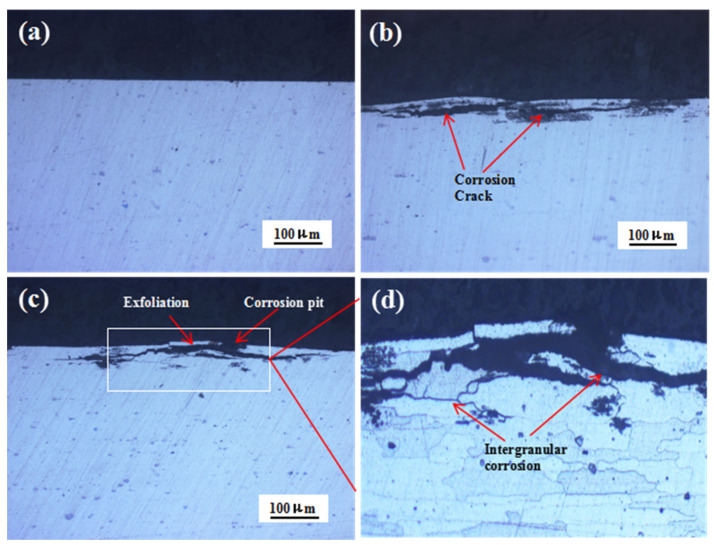
Cross-sectional morphologies of the 7075 alloy after outdoor exposure to the marine atmospheric environment at different time points: (**a**) 0 months, (**b**) 6 months, and (**c**) 12 months. (**d**) Magnified image of the white rectangle in (**c**).

**Figure 6 materials-16-02396-f006:**
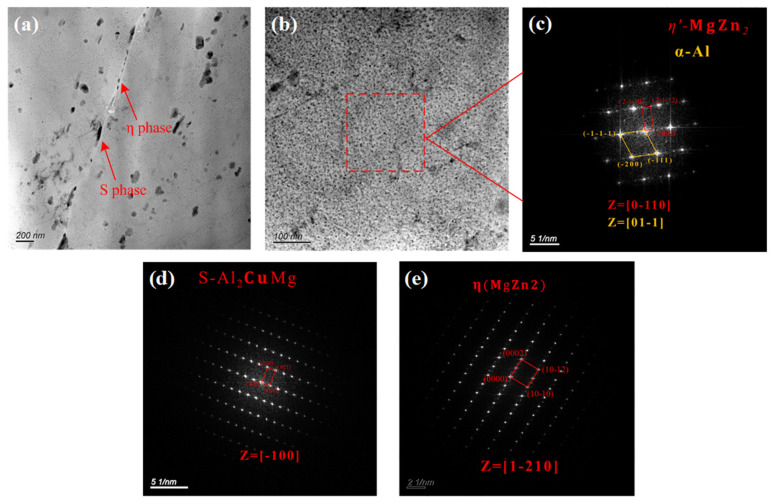
TEM images of the secondary phases of the 7075 alloy: (**a**) bright-field image of the grain boundary; (**b**) bright-field image of the grain interior; (**c**) diffraction spots of the secondary phase in the grain; (**d**) diffraction spots of S phase along the grain boundary; and (**e**) diffraction spots of η phase along the grain boundary.

**Figure 7 materials-16-02396-f007:**
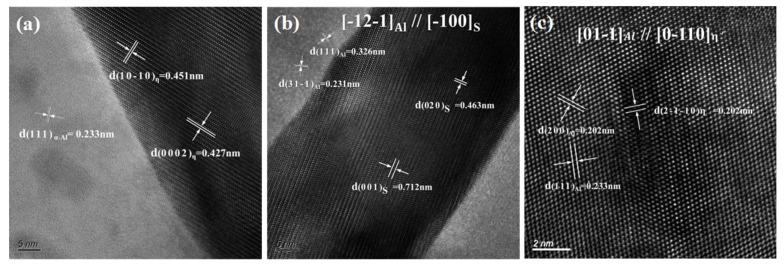
High-resolution images of the secondary phases: (**a**) η(MgZn_2_) phase and (**b**) S(Al_2_CuMg) in the grain boundary; (**c**) $\eta^{\prime }$ phase in the grain.

**Figure 8 materials-16-02396-f008:**
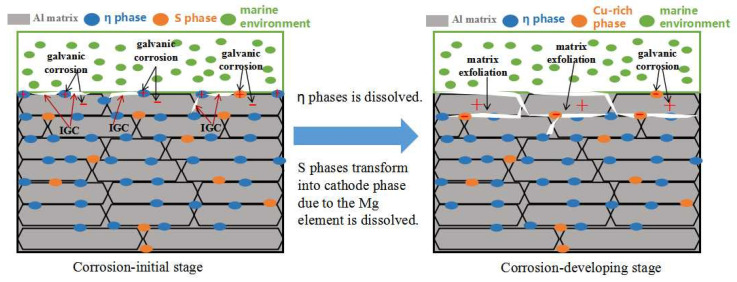
Sketch of the corrosion processes of the 7075 alloy’s exposure to the marine environment.

**Figure 9 materials-16-02396-f009:**
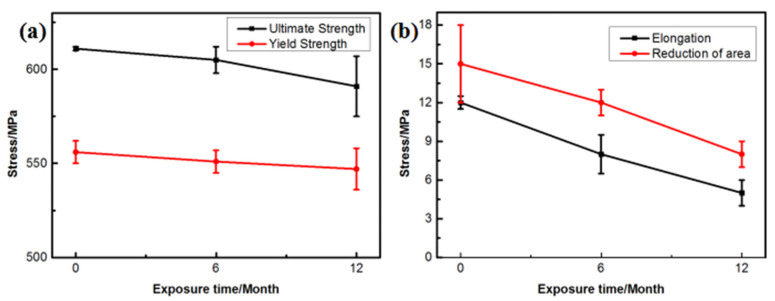
Tensile properties of the 7075 alloy exposed to the marine atmospheric environment at different time points: (**a**) strength and (**b**) plasticity.

**Figure 10 materials-16-02396-f010:**
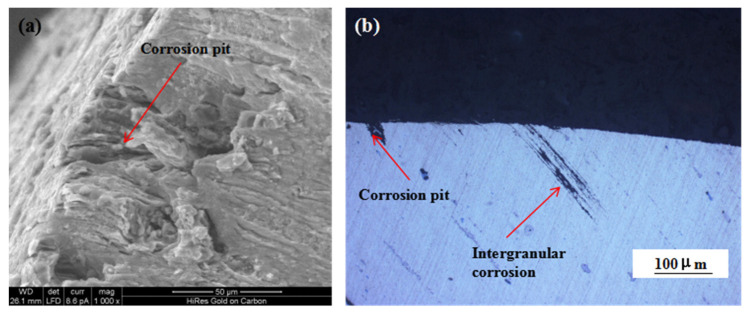
Fracture (**a**) and cross-sectional (**b**) morphologies of the 7075 alloy tensile specimen after exposure to the marine atmospheric environment.

**Table 1 materials-16-02396-t001:** Composition of the 7075 aluminum alloy (wt.%).

Element	Cu	Fe	Mg	Zn	Cr	Ti	Si	Mn
Content	1.51	0.14	2.58	5.57	0.2	0.024	<0.05	0.045

## Data Availability

Not applicable.

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
