# Peer review of "Impact of Marine Atmospheric Corrosion on the Microstructure and Tensile Properties of 7075 High-Strength Aluminum Alloy"

_materials, 2023, doi:10.3390/ma16062396_

Round 1

Reviewer 1 Report

I have had the opportunity to read your manuscript entitled "Impact of Marine Atmospheric Corrosion on the Microstructure and Tensile Properties of the 7075 High-Strength Aluminum Alloy". As a reviewer, I would like to offer some comments and suggestions that I believe would enhance the quality of your work.

Firstly, I would like to note that pitting and intergranular corrosion are common types of corrosion reported for aluminum and its alloys when exposed to marine environments or simulated seawater. As such, I am concerned that this study may lack novelty. Upon reviewing the literature, I found that you had previously reported on the pre-corrosion fatigue performance of the 7075 alloy in an actual marine environment [1] and you have found similar results in terms of corrosion. However, I do believe that your study's focus on defects due to corrosion and their impact on the mechanical properties (tensile and plasticity properties) of the aluminum alloy 7075 offers a degree of originality. Nonetheless, I suggest that you expand on this result and provide further insight into the mechanisms responsible for the observed effects.

Lastly, I would like to address some additional details that should be considered.

1.  Could you please indicate the data source for the chemical composition of the aluminum alloy?

2. I would like to inquire about the meaning of the letter "a" used to denote the corrosion rate in g/m2 a. It may be clearer to use SI units, such as g m-2 d-1 (d=day).

3. I noticed that there are variations in the page formatting of your reference list. I recommend that you format your references homogeneously to improve the clarity and consistency of your work.

4. I would like to recommend an amplification of the region with the peak corresponding to Al(OH)3 to clarify the conflict with the corresponding conclusion. It would also be helpful to include the reference data, such as the corresponding JCPDS cards.

Thank you for considering my comments, and I look forward to seeing the revised manuscript.

Reviewer 2 Report

The authors presented results of the impact of marine atmospheric corrosion (in natural conditions) on the microstructure of the aluminum alloy. 

The manuscript is well organized and experiments as well as presentation of the results are clearly presented.

Before publishing some corrections, in order to improve the quality of the presentation, should be done. ?

In the Abstract the authors must correct the first sentence !..study aimed to study..."

Figure 4 Caption must contain information concerning exposure time.

In order to obtain more clear inside of the corrosion mechanism, the corrosion mechanism (page 7) should be presented as Scheme with boundaries and processes that occur in each phase and boundary.

Reviewer 3 Report

Dear Authors, 

I have read your manuscript proposed for publication in the journal Materials. The subject matter covered is quite interesting, however, the quality of the work is very poor. 

Basically, there is no analysis of the state of the art in the subject matter covered. The methodology and its description has huge gaps, e.g:

- no explanation of why this heat treatment status of the analysed alloy was chosen, do the different processes of this treatment affect the susceptibility to corrosion? 

- no information on the process of making samples from the alloy;

- no information on which direction relative to the rolling direction the samples were prepared from; 

- no information if and how residual stresses after the rolling process were taken into account? 

- no information on how the chemical composition of the analysed alloy was determined? 

- no information on changes in temperature, humidity, precipitation, wind during the test period? 

- no information on average salinity of sea water and salinity of the air in analysed localisation,

etc. 

The methodology mentions measuring the weight loss of the samples, while no such results are presented in the results. 

The article needs to be rewritten and completed, then only then can it be resubmitted.

Reviewer 4 Report

The research article is about the effect of the marine atmosphere on corrosion and microstructure as well as the mechanical properties of 7075 Al alloy. The novelty of the research is well explained in the manuscript. The following may be added to improve the quality of the article.

1. Temperature variation during the exposure time of the samples can be added. In the marine environment, the temperature effect can not be neglected.

2. In the figure 2 (a) and (b), the phases and composition can be shown.

3. In section 3.2, a justification may be added for the fast corrosion of 7075 as compared to 7085 alloys for which earlier studies have been carried out.  

Round 2

Reviewer 3 Report

Dear Authors,

your manuscript still does not include correctly introduction. Chapter 3 " Results and Discussion" includes only results without discussion in relation to other literature sources. 

If you determined chemical conteins of material you should describe the method and used devices for this process. 

You didn't descibed machining methods and cutting tools used to samples preparation process. Material of cutting tools can affected on machined material corrosion. 
